Characterization of the bark storage protein gene (JcBSP) family in the perennial woody plant Jatropha curcas and the function of JcBSP1 in Arabidopsis thaliana

Zhang Ming-Jun 1 2
Fu Qiantang 2
Chen Mao-Sheng 2
He Huiying 2
Tang Mingyong 2
Ni Jun 3
Tao Yan-Bin taoyanbin@xtbg.ac.cn 2
Xu Zeng-Fu zfxu@gxu.edu.cn 2 3
1 School of Life Sciences, University of Science and Technology of China, Hefei , Anhui , China
2 CAS Key Laboratory of Tropical Plant Resources and Sustainable Use, Xishuangbanna Tropical Botanical Garden, The Innovative Academy of Seed Design, Chinese Academy of Sciences, Menglun, Mengla , Yunnan , China
3 State Key Laboratory for Conservation and Utilization of Subtropical Agro-Bioresources, College of Forestry, Guangxi University, Nanning , Guangxi , China
Ma Wei
Electronic publication date: 2022 Feb 8
Publication date: 2022
Volume: 10
Electronic Location ID: e12938
Received 2021 Nov 10; Accepted 2022 Jan 24
Copyright: ©2022 Zhang et al.
Copyright year: 2022
Copyright holder: Zhang et al.
License: This is an open access article distributed under the terms of the Creative Commons Attribution License, which permits unrestricted use, distribution, reproduction and adaptation in any medium and for any purpose provided that it is properly attributed. For attribution, the original author(s), title, publication source (PeerJ) and either DOI or URL of the article must be cited.
License URL: https://creativecommons.org/licenses/by/4.0/

Keywords: Jatropha curcas, JcBSP gene family, Seasonal nitrogen cycling, Nitrogen induction, Overexpression, Arabidopsis thaliana

Funding: The National Natural Science Foundation of China No. 31771605 The Chinese Academy of Sciences No. kfj-brsn-2018-6-008 This work was supported by funding from the National Natural Science Foundation of China (No. 31771605) and the Chinese Academy of Sciences (No. kfj-brsn-2018-6-008). The funders had no role in study design, data collection and analysis, decision to publish, or preparation of the manuscript.

==============================
Background

Bark storage protein (BSP) plays an important role in seasonal nitrogen cycling in perennial deciduous trees. However, there is no report on the function of BSP in the perennial woody oil plant Jatropha curcas.

Methods

In this study, we identified six members of JcBSP gene family in J. curcas genome. The patterns, seasonal changes, and responses to nitrogen treatment in gene expression of JcBSPs were detected by quantitative reverse transcription-polymerase chain reaction (qRT-PCR). Overexpression of JcBSP1 in transgenic Arabidopsis thaliana was driven by a constitutive cauliflower mosaic virus (CaMV) 35S RNA promoter.

Results

JcBSP members were found to be expressed in various tissues, except seeds. The seasonal changes in the total protein concentration and JcBSP1 expression in the stems of J. curcas were positively correlated, as both increased in autumn and winter and decreased in spring and summer. In addition, the JcBSP1 expression in J. curcas seedlings treated with different concentrations of an NH4NO3 solution was positively correlated with the NH4NO3 concentration and application duration. Furthermore, JcBSP1 overexpression in Arabidopsis resulted in a phenotype of enlarged rosette leaves, flowers, and seeds, and significantly increased the seed weight and yield in transgenic plants.

Introduction

Seasonal nitrogen cycling (SNC) is important in deciduous perennials to ensure the sufficient use of nitrogen resources. It is also a decisive factor for plant fitness in perennial (May & Killingbeck, 1992). The process of SNC involves the degradation of proteins when leaves shed in autumn, the transportation of the released amino acids to the perennial tissues (bark and wood) to synthesize storage proteins, and then used for the growth of new stems and leaves in spring (Babst & Coleman, 2018; Wildhagen et al., 2010). When the deciduous perennial trees overwinter, nitrogen is transported from senescing leaves to perennial tissues for storage (Ryan & Bormann, 1982). For example, before poplar leaf senescence, protein is hydrolyzed and transported to stems and roots in the form of amino acids, which results in approximately 90% of the nitrogen being removed from the leaves (Chapin & Kedrowski, 1983; Pregitzer et al., 1990). Through longitudinal section observation of Populus stems in winter and summer, it has been found that the phloem parenchyma cells and xylem ray cells contained only a large central vacuole in summer, while the central vacuole in these cells was replaced by many small protein storage vacuoles in winter (Clausen & Apel, 1991; Cleve & Apel, 1993; Cleve, Clausen & Sauter, 1988; Cooke & Weih, 2005; Sauter, Cleve & Wellenkamp, 1989; Sauter & Cleve, 1992; Wetzel, Demmers & Greenwood, 1989a). This protein is a kind of vegetative storage protein (VSP) and is a designated bark storage protein (BSP) (Cooke & Weih, 2005). It is an important form of nitrogen storage for perennial woody plants in winter.

Previous studies have shown that the poplar BSP is composed of a multigene family, including three subfamilies: BSP, wound-inducible 4 (WIN4), and poplar nitrogen-inducible 288 (PNI288) (Coleman, Chen & Fuchigami, 1992; Wildhagen et al., 2010). All three subfamily genes can respond to nitrogen induction (Coleman, Baíiados & Chen, 1994; Lawrence et al., 2001; Lawrence et al., 1997), but only BSP has been found to exhibit consistent seasonal expression changes with the total protein concentration in bark. The expression of BSP was increased in winter, whereas the expression of WIN4 and PNI288 increased only in spring (Wildhagen et al., 2010), and BSP has been reported to respond to short light duration and low temperature induction (Black et al., 2001; Cleve & Apel, 1993; Coleman et al., 1991; Coleman et al., 1993; Lawrence et al., 2001; Wildhagen, Bilela & Rennenberg, 2013). Therefore, only BSP is directly related to nitrogen storage during plant dormancy in winter among these three subfamilies.

Jatropha curcas is a perennial woody oil plant of the Euphorbiaceae family. It has received widespread attention because its seed oil is recognized as a promising feedstock for biodiesel production (Divakara et al., 2010; Kamel et al., 2018; Makkar & Becker, 2009; Mazumdar et al., 2018; Mohibbeazam, Waris & Nahar, 2005; Pandeya et al. 2012; Pramanik, 2003; Vaknin et al., 2018; Yi et al., 2014). Although J. curcas grows in tropical and subtropical regions, it is also a deciduous tree. Adult J. curcas begins to defoliate in autumn and stays dormant in winter until the next spring, when it enters the growing season. In this study, to identify J. curcas BSP (JcBSP) genes that may be involved in seasonal nitrogen cycling, we examined the expression of JcBSP family members in response to seasonal changes and nitrogen induction and found that the expression of JcBSP1 was positively correlated with the total protein concentration in the stems during seasonal changes and with the exogenous nitrogen application. To further determine the roles of JcBSP1 in plant growth and development, we characterized phenotypic changes in transgenic Arabidopsis thaliana overexpressing JcBSP1 and found that transgenic plants exhibited phenotypes of enlarged rosette leaves, flowers, and seeds. These findings laid the foundation for further research on the function of BSP genes in the plant growth and development.

Materials & Methods

Plant materials and nitrogen treatment

Four-year-old adult J. curcas were grown in the experimental field of Xishuangbanna Tropical Botanical Garden (21°54′N, 101°46′E; 580 m in altitude) in Yunnan Province. Wild-type Arabidopsis thaliana ecotype Columbia (Col-0) and the transgenic lines were grown in an environmentally controlled room at 22 °C under a 16-h light/8-h dark photoperiod. Wild-type J. curcas seeds were sown in sand that had been washed several times with distilled water and grown at 30 °C under a 12-h light/12-h dark photoperiod. Then, two-month-old J. curcas seedlings were randomly divided into three groups, with 12 plants per group. The three groups were treated with different concentrations of NH4NO3 solution (0, 5, and 50 mM). The seedlings were watered every week with 100 mL of a NH4NO3 solution per cup of seedlings.

Sequence analysis

We used BLAST (http://www.ncbi.nlm.nih.gov/BLAST/) to analyze the cDNA sequence, CDS and amino acid sequence of JcBSP gene family members in the NCBI database. The conserved domains of deduced protein sequences were analyzed by the NCBI Conserved Domain Database (NCBI-CDD, https://www.ncbi.nlm.nih.gov/Structure/cdd/wrpsb.cgi). The alignment of amino acid sequences was performed using DNAMAN software (version 6, Lynnon Biosoft Corporation, Canada, https://www.lynnon.com/dnaman.html).

Phylogenetic analysis

To examine the phylogenetic relationships of the BSP homologues from different species, we retrieved the deduced protein sequences from the NCBI database (https://www.ncbi.nlm.nih.gov/) and selected the sequences from species belonging to Euphorbiaceae and Populus with the highest sequence similarity to JcBSPs. Sequences of PtdPNI288 and PtdWIN4 were derived from the reference literature (Cooke & Weih, 2005; Wildhagen et al., 2010). A phylogenetic tree was built in the MEGA program (version 7.0, https://megasoftware.net/) using the neighbor-joining method with 1000 bootstrap replicates.

RNA extraction

The samples for RNA extraction included different tissues (roots, stems, shoot apices, young leaves, mature leaves, male flowers, female flowers, fruits and seeds) of four-year-old adult J. curcas, stems collected from October 2019 to August 2020, two-month-old J. curcas seedlings treated with different concentrations of NH4NO3 solution for 0, 2, 4, 6, and 8 weeks, and leaves of one-month-old WT and transgenic Arabidopsis. These samples were quickly frozen in liquid nitrogen and stored at −80 °C. Total RNA was extracted using the silica adsorption method (Ding et al., 2008) , and the concentration and purity of RNA were detected by spectrophotometry and agarose gel electrophoresis, respectively.

qRT-PCR analysis

Reverse transcription of total RNA was performed using the PrimeScript® RT reagent kit with gDNA Eraser (Takara, Dalian, China). qRT-PCR was performed on the Roche 480 real-time PCR detection system using LightCycler® 480 SYBR Green I Master Mix (Roche Diagnostics, Indianapolis, IN, USA). The qRT-PCR reactions were performed under the following conditions: 5 min at 95 °C for the initial denaturation, followed by 42 cycles of 10 s at 95 °C, 20 s at 57 °C, and 20 s at 72 °C for the PCR amplification, and 1 cycle of 30 s at 95 °C, 30 s at 65 °C and 0.06 °C/s heating up to 95 °C for the melting curve. Data was analyzed using the 2−ΔΔCT method as described by Livak & Schmittgen (2001). All expression data obtained in the qRT-PCR assay were normalized to the expression of JcActin1 (Zhang et al., 2013) and AtActin2. The primers used for qRT-PCR are listed in Table S1.

Protein extraction

Stem samples of four-year-old adult J. curcas collected from October 2019 to August 2020 were also used for total protein extraction. The bark + phloem and xylem + pith dissected from stems were ground into powder mixtures. Fifty milligrams of powder were homogenized in 300 µl of protein extract buffer (50 mM Tris-HCl pH 7.4, 150 mM NaCl, 1 mM EDTA, 0.1% Triton X-100, 10% glycerol). The mixture was incubated at 4 °C for 30 min and then centrifuged at 12,000 rpm for 15 min at room temperature. The supernatant was collected and analyzed by a Bradford protein assay kit (BL524A, GUANGKE Technology Company, Kunming).

Correlation analysis

Correlation analysis between the total protein concentration and JcBSP1 expression was performed by the method of Spearman’s rank correlation using R package ggpubr (version 0.4.0, https://cran.r-project.org/).

Construction of the JcBSP1 overexpression vector and Arabidopsis transformation

The primers XD423 (CGAGCTCATGGCTATGGCGACGGTGAT) and XD424 (CGGATCCTCACTCTTCATCATAACGGA) carrying SacI and BamHI restriction sites, respectively, were used to clone the full-length JcBSP1 CDS. Then, the PCR product was cloned into the pGEM-T Easy vector (Promega, Madison, WI, USA). To generate the 35S:JcBSP1 overexpression vector, SacI and BamHI were used to digest the plant transformation vector pOCA30 (Chen & Chen, 2002) and the pGEM-T Easy vector containing the JcBSP1 sequence, respectively, and then, the resulting fragments were ligated by using T4 DNA Ligase (Promega). The generated 35S:JcBSP1 plasmid was transferred to Agrobacterium tumefaciens EHA105. Transformation of Arabidopsis was performed using the floral dip method (Clough & Bent, 1998).

Results

Identification of the members of the JcBSP gene family

We found six members of the JcBSP gene family in J. curcas from the NCBI database using BLAST and designated them as JcBSP1, JcBSP2, JcBSP3, JcBSP4, JcBSP5, and JcBSP6 (Table 1). All the JcBSP gene family members contain a conserved domain, purine nucleoside phosphorylase_uridine phosphorylase_1 (PNP_UDP_1), which is a signature of the phosphorylase superfamily (Fig. 1).

Table 1 Sequence information for members of the JcBSP family in J. curcas.

Gene
name	GenBank
accession number	cDNA
(bp)	CDS
(bp)	Number of
amino acids (aa)	
JcBSP1	XM_012218517	1,248	972	323	
JcBSP2	XM_012214829	1,144	1,017	338	
JcBSP3	XM_012218526	959	843	280	
JcBSP4	XM_012218520	1,038	702	233	
JcBSP5	XM_012222025	1,274	1,062	353	
JcBSP6	XM_012222124	1,174	1,041	346	

Figure 1 Protein sequence alignment of JcBSP family members of J. curcas.

Identically conserved amino acid sequences are shown with a dark blue background, and partially conserved amino acid sequences are shown with grey and brown backgrounds; the conserved PNP_UDP_1 domain of JcBSP is indicated with overlining.

To investigate the evolutionary relationships among BSP homologous genes, we performed phylogenetic analysis of genes from J. curcas and other species. The phylogenetic tree showed that JcBSP1, JcBSP5, and JcBSP6 were closely related to the BSP homologues from Euphorbiaceous species, and JcBSP2, JcBSP3, and JcBSP4 were closely related to the BSP homologues from poplar (Fig. 2).

Figure 2 Phylogenetic tree analysis of BSP homologues.

The homologues compared with J. curcas BSPs include Ricinus communis RcBSP; Populus trichocarpa PtBSP; Populus deltoids PdBSP; Populus alba PaBSP and PaWIN4; Hevea brasiliensis HbBSP; Manihot esculenta MeBSP; P. trichocarpa × P. deltoids PtdWIN4 and PtdPNI288; and P. trichocarpa × P. alba PtaWIN4 and PtaPNI288. The phylogenetic tree was constructed by the neighbor-joining method in MEGA 7.0 software; one thousand replicates were used for the bootstrap test; red frame: JcBSP family members.

The expression patterns of JcBSPs in J. curcas

To analyze the expression patterns of JcBSP family members, we used qRT-PCR to detect the expression levels of JcBSPs in roots, stems, shoot apices, young leaves, mature leaves, male flowers, female flowers, fruits, and seeds of adult J. curcas. The results showed that JcBSP1 and JcBSP2 exhibited similar expression patterns, in which both were highly expressed in shoot apices, stems, young leaves and female flowers; JcBSP3 was mainly expressed in roots, stems, shoot apices, male flowers and fruits; the expression of JcBSP4 was concentrated in reproductive tissues, with the highest expression level in female flowers; the expression of JcBSP5 was concentrated in vegetative tissues, with the highest expression level in stems; and JcBSP6 was remarkably expressed in male flowers (Fig. 3). Based on these results, we hypothesized that JcBSPs could play important roles in the growth and development of various organs, except seeds, in which all the members were barely expressed. This finding also indicates that JcBSPs may be vegetative storage proteins rather than seed storage proteins.

Figure 3 Expression analysis of JcBSPs in various tissues of adult J. curcas.

The qRT-PCR results were obtained from three biological replicates and three technical replicates. The values were normalized to the expression of JcActin1. Error bars denote the standard deviation (SD) calculated from three biological replicates. R, roots; S, stems; SA, shoot apices; YL, young leaves; ML, mature leaves; MF, male flowers; FF, female flowers; Fr, fruits; Se, seeds.

Seasonal changes in total protein concentrations and JcBSP expression in the stems of J. curcas

In perennial deciduous trees, most nitrogen resources in senescing leaves are transported to perennial tissues (bark and wood), and stored as proteins during autumn and winter; the next spring, these proteins are hydrolyzed to amino acids, which are transported from perennial tissues to growing tissues (Chapin & Kedrowski, 1983; Cooke & Weih, 2005; Sauter, Cleve & Wellenkamp, 1989). Therefore, we investigated whether the total protein concentration in J. curcas stems was relevant to seasons.

From October 2019 to August 2020, we sampled the stems of adult J. curcas in two parts (bark + phloem and xylem + pith), as shown in Fig. 4A, and examined the total protein concentrations of the samples. The results showed that the total protein concentrations in the bark + phloem and xylem + pith were approximately 10.5 mg/g FW in October; then, J. curcas entered the dormant stage in December, and the total protein concentrations reached a peak in the bark + phloem (12.3 mg/g FW) and xylem + pith (16.4 mg/g FW). When J. curcas began to enter the growing season in March, the total protein concentrations decreased sharply to 7 mg/g FW in the bark + phloem, which further decreased to 5.5 mg/g FW in August. The total protein concentration in the xylem + pith showed a similar trend (Fig. 4B). The total protein concentration in the J. curcas stem exhibited a seasonal change, which accumulated in autumn and winter and decreased in spring and summer. This result indicates that the total protein in the stems is a form of nitrogen storage during the overwintering period of J. curcas, and this protein is reallocated during the growing seasons.

Figure 4 Seasonal changes in total protein concentrations in the stems of adult J. curcas.

(A) Cross section of stems. The red arrow shows the bark and phloem, and the blue arrow shows the xylem and pith. (B) Total protein concentrations of J. curcas stems. The results were obtained from three biological replicates. Error bars denote the SD calculated from three biological replicates.

Based on the above results, we examined the seasonal course of JcBSP expression in the same samples mentioned above (Fig. 4A) to investigate whether the expression of JcBSPs shows the same seasonal changes as the total protein concentration. The results showed that the expression of JcBSP family members in stems exhibited different patterns over the seasonal course (Fig. 5). From autumn to winter, the expression of JcBSP1 in the two parts of the stems increased rapidly, with a higher level in xylem + pith, and then decreased sharply in spring and remained low until August. The seasonal changes in JcBSP1 expression in the two parts of the stem were entirely consistent with those of the total protein concentration. However, the seasonal expression patterns of JcBSP2, JcBSP4 and JcBSP5 in stems were inconsistent with those of the total protein concentration. In addition, only in the xylem + pith did the expression of JcBSP3 and JcBSP6 show the same seasonal changes as that of the total protein concentration, but their expression levels were much lower than that of JcBSP1. Therefore, we analyzed the correlation between seasonal changes in the total protein concentration and JcBSP1 expression. It turned out that there were significant correlations between them in the bark + phloem (r = 0.63, P < 0.01) and the xylem + pith (r = 0.67, P < 0.01) (Fig. 6). These results suggest that JcBSP1 might play an important role in seasonal nitrogen cycling.

Figure 5 Seasonal changes in JcBSP expression in the two parts of J. curcas stems.

The qRT-PCR results were obtained from three biological replicates and three technical replicates. The values were normalized to the expression of JcActin1. Error bars denote the SD calculated from three biological replicates.

Figure 6 Correlation analysis between the seasonal changes in total protein concentration and JcBSP1 expression in the bark + phloem (A) and the xylem + pith (B).

JcBSP expression in response to nitrogen

To further verify the correlation between JcBSPs and seasonal nitrogen cycling, we investigated the response of these genes to nitrogen induction. We applied a 0, 5, and 50 mM NH4NO3 solution to two-month-old J. curcas seedlings. After 8 weeks of treatment, we found that the group treated with the 5 mM NH4NO3 solution grew better than the other two groups (Fig. 7A). This result indicated that nitrogen supply in a certain concentration range could effectively promote the growth of J. curcas.

Figure 7 Changes in JcBSP expression in the leaves of two-month-old J. curcas seedlings treated with NH4NO3.

(A) J. curcas seedlings treated with different concentrations of NH4NO3 for 8 weeks. (B)JcBSP expression in response to NH4NO3 treatment. The qRT-PCR results were obtained from three biological replicates and three technical replicates. The values were normalized to the expression of JcActin1. Error bars denote the SD calculated from three biological replicates.

We collected the leaves of J. curcas seedlings treated with different concentrations of NH4NO3 for 0, 2, 4, 6, and 8 weeks to detect changes in JcBSP family member expression. The results showed that the expression of JcBSP1 increased obviously along with the increased NH4NO3 concentration and application duration; JcBSP2 expression increased remarkably with the 5 mM NH4NO3 treatment for 2–8 weeks and with 50 mM NH4NO3 treatments for 8 weeks; JcBSP4 expression was induced obviously with only the 50 mM NH4NO3 treatment for 4 weeks; however, the expression of JcBSP3, JcBSP5 and JcBSP6 was not induced by NH4NO3 treatment (Fig. 7B). The results indicated that JcBSP1 and JcBSP2 expression is responsive to nitrogen induction. In particular, JcBSP1 expression was positively correlated with the nitrogen concentration and application duration. Combined with the seasonal changes in JcBSP1 expression in stems, we further concluded that JcBSP1 might be a form of nitrogen storage in the seasonal nitrogen cycling in J. curcas.

Overexpression of J. curcas JcBSP1 in Arabidopsis led to enlarged rosette leaves, flowers, and seeds

Next, we investigated the function of JcBSP1 in plant growth and development in transgenic Arabidopsis. Twenty-two independent 35S:JcBSP1 transgenic Arabidopsis lines were generated (Fig. 8A). JcBSP1 expression in seven transgenic lines showing similar phenotypes was analyzed, and most of transgenic lines yielded abundant transgene expression (Fig. S1). We investigated in further detail the phenotypes of two independent transgenic lines, L4 and L10, which exhibited high and intermediate expression levels of JcBSP1, respectively (Fig. 8B).

Figure 8 Phenotypic changes in 35S:JcBSP1 transgenic Arabidopsis.

(A) PCR identification of 35S:JcBSP1 transgenic Arabidopsis. WT, wild-type negative control; ddH2O, ddH2O negative control; P, positive control. (B) qRT-PCR analysis of JcBSP1 expression in WT and transgenic Arabidopsis (L4 and L10). The qRT-PCR results were obtained from three biological replicates and three technical replicates. The values were normalized to the expression of AtActin2. Error bars denote the SD calculated from three biological replicates. (C and D) Rosette leaves and flowers from WT and transgenic Arabidopsis. (E and F) Leaf length and width of WT and transgenic Arabidopsis. The values are presented as the means ± standard deviations (n = 8). Student’s t-test was used for significance analysis: **P ≤ 0.01.

During plant growth and development, we found that 35S:JcBSP1 transgenic Arabidopsis produced larger rosette leaves and flowers than the wild-type (WT) plants (Figs. 8C and 8D). As shown in Figs. 8E and 8F, the rosette leaf lengths and widths were all significantly increased in transgenic lines. And larger seeds and significantly increased hundred-seed weights were found in transgenic plants (Figs. 9A and 9B). Consequently, the seed yields in transgenic lines were significantly higher than that of the WT (Fig. 9C). These results indicated that JcBSP1 was able to affect plant growth and development.

Figure 9 Overexpression of JcBSP1 increased the size, weight, and yield of seeds in transgenic Arabidopsis.

Seed size (A), hundred-seed weight (B) (n = 3) and seed yield per plant (C) (n = 6) were analyzed in WT and transgenic Arabidopsis lines L4 and L10. The values are presented as the means ± standard deviations. Student’s t-test was used for significance analysis: *P ≤ 0.05, **P ≤ 0.01.

Discussion

BSP, a kind of VSP, is a main nutrient storage protein in perennial woody plants and a form of nitrogen storage in vegetative tissues (Cooke & Weih, 2005; Staswick, 1994). It is different from seed storage protein, which accumulates during seed maturation and provides a nitrogen source for embryo development (Autran, Halford & Shewry, 2001; Gacek, Bartkowiak-Broda & Batley, 2018; Kawakatsu et al., 2010). In Populus, BSP has been found to be highly expressed in the bark, dormant cambium and bud (Coleman, Baíiados & Chen, 1994; Cooke & Weih, 2005) , and the bspA promoter has been shown to be predominantly active in bark (Zhu & Coleman, 2001). In this study, we identified six members of the JcBSP gene family in J. curcas, and none of them were expressed in seeds (Fig. 3), indicating that JcBSP might be a nutrient storage protein rather than seed storage protein. In addition, JcBSP1, JcBSP2 and JcBSP4 were highly expressed in female flowers, and JcBSP3 and JcBSP6 were relatively highly expressed in male flowers, suggesting that they may be involved in the development of female and male flowers, respectively.

In perennial woody plants, BSP plays an important role in seasonal nitrogen cycling (Wetzel, Demmers & Greenwood, 1989b; Wetzel & Greenwood, 1989; Wildhagen et al., 2010). During autumn and winter, nitrogen-rich amino acids are transported from senescing leaves to perennial tissues and subsequently used to synthesize proteins for nitrogen storage (Geßler, Kopriva & Rennenberg, 2004; Hörtensteiner & Feller, 2002). BSP is the main form of nitrogen storage in trees during the dormant period, which accumulates in autumn and winter (Cooke & Weih, 2005; Wetzel, Demmers & Greenwood, 1989b). In this study, we found that the seasonal changes in JcBSP1 expression in stems were consistent with those of the total protein concentration, as both increased in autumn and winter and then decreased in spring and summer (Figs. 4B and 5). And there is a significant correlation between seasonal changes in the total protein concentration and JcBSP1 expression (Fig. 6), suggesting that JcBSP1 might be the main protein stored in the stem of J. curcas during overwintering. Moreover, the expression of JcBSP1 was positively correlated with the nitrogen concentration and application duration (Fig. 7B). Therefore, we hypothesized that JcBSP1 might play an important role in the seasonal nitrogen cycling of J. curcas, acting as a form of nitrogen storage in the stems during overwintering. In addition, this study was conducted in the Xishuangbanna Tropical Botanical Garden, Chinese Academy of Sciences, which is located in a tropical region of China. According to rainfall, there are two seasons, a rainy season from May to October and a dry season from November to April of the following year, in Xishuangbanna area. The dry season is further divided into the foggy-cool season from November to February of the following year and the dry-hot season from March to April. Although the foggy-cool season has little precipitation, there is a large amount of dense fog from night to noon, which has a certain compensation effect on the water demand of plants in dry season; the dry-hot season has a dry climate, low precipitation and large daily temperature differences (Zhao et al., 2009). As shown in Figs. 4B and 5, both of the total protein concentration and JcBSP1 expression in the stems were decreased to a very low level at the beginning of the dry-hot season. Hence, further studies are required to link the seasonal changes in total protein concentration and JcBSP1 expression to possible drought-related protein mobilization.

It is well known that nitrogen is an important nutrient for plant growth and development. Lemaitre et al. (2008) showed that when Arabidopsis grew under low nitrogen conditions, rosette biomass and seed yield were limited. Storage proteins are considered as nitrogen source that are utilized for plant growth (Sözen, 2004; Titus & Kang, 1982). In this study, we found that overexpression of JcBSP1 could promote the growth and development of rosette leaves, flowers, and seeds in transgenic Arabidopsis (Figs. 8 and 9). This finding further indicates the JcBSP1 might be a form of nitrogen storage in plants, serving as a nutrient provider. Similarly, overexpression of a storage protein gene AmA1 in potato could increase the growth and production of tubers (Chakraborty, Chakraborty & Datta, 2000). By analyzing cell architectures, the cell areas in cortex, perimedullary and pith regions of the tuber were found to be increased, which indicated the AmA1 storage protein in potato tuber was correlated with cell growth (Agrawal et al., 2013). In cabbage, when the nitrogen supply can’t meet the need of plant growth, the leaf cells became smaller while the number of cell layers remained unchanged (Kano et al., 2007). It turns out that both endogenous and exogenous nitrogen sources could affect cell growth. Accordingly, overexpression of JcBSP1 in transgenic Arabidopsis may also promote the cell growth, resulting in enhanced plant growth and production. Furthermore, it is worthy to mention here that although the expression level of JcBSP1 in the transgenic line L4 was higher than that in L10 (Fig. 8B), the leaves and seeds in L4 were relatively smaller than those in L10 (Figs. 8C, 8E, 8F; 9). We hypothesized that this phenotype might be caused by the excessively high JcBSP1 transgene expression, which might lead to excess JcBSP1 protein storage and subsequently excess nitrogen accumulation in L4 plants. Previous study showed that under the excess nitrogen conditions, both cell number and size were found to be reduced in leaves (MacAdam, Volenec & Nelson, 1988). In addition, about half of the Rubisco are inactive or only half of the catalytic sites are functional, which certainly leads to a decrease in photosynthetic efficiency and therefore a retardation in plant growth (Chapin, Schulze & Mooney, 1990; Cheng & Fuchigami, 2000; Millard, 1988). As shown in Fig. 7A, the excessive nitrogen supply does have a certain negative impact on J. curcas growth. Consistently, Barbosa et al. (2010) also found that when the adding nitrogen concentration was below 40 mM, it stimulated Arabidopsis root growth, while the concentration was higher than 40 mM, root elongation was inhibited.

In addition, VSPs may also play a role in plant defense. In Arabidopsis, AtVSP1 and AtVSP2 have been shown to enhance plant resistance to diseases and insects (Berger, Mitchell-Oldsb & Stotz, 2002; Ellis & Turner, 2001; Liu et al., 2005). Furthermore, AtVSP1 and AtVSP2 have been found to be highly expressed in flowers (Utsugi et al., 1998), which implies a mechanism used by Arabidopsis to protect reproductive structures (Liu et al., 2005). Interestingly, most JcBSPs were also highly expressed in female or male flowers (Fig. 3). Thus, in addition to being a provider of nitrogen resources, JcBSPs may also play other roles in plant growth and development, which requires further study.

Conclusions

In this study, six members of the JcBSP gene family were identified in J. curcas, which were expressed in various tissues, except seeds. Among these members, only the expression of JcBSP1 was positively correlated with the total protein concentration in the stems during seasonal changes and with the exogenous nitrogen application. We thus supposed that JcBSP1 could play an important role in seasonal nitrogen cycling as a form of nitrogen storage. By the function analysis of JcBSP1 in transgenic Arabidopsis, we found that JcBSP1 was able to enhance the plant growth and production. This suggests that JcBSP1 could be useful in crop breeding.

Supplemental Information

Supplemental Information 1 QRT-PCR primers list

Click here for additional data file.

Supplemental Information 2 JcBSP1 expression in wild-type (WT) and transgenic Arabidopsis lines

The qRT-PCR results were obtained from three biological replicates and three technical replicates. The values were normalized to the expression of AtActin2. Error bars denote the SD from three biological replicates.

Click here for additional data file.

Supplemental Information 3 Seasonal changes in JcBSP expression in the stems

Click here for additional data file.

Supplemental Information 4 JcBSP expression in response to nitrogen

Click here for additional data file.

Supplemental Information 5 JcBSPs expression patterns

Click here for additional data file.

Supplemental Information 6 Seasonal changes in total protein concentrations in the stems

Click here for additional data file.

Supplemental Information 7 Leaf length and width

Click here for additional data file.

Supplemental Information 8 Hundred-seed weight

Click here for additional data file.

Supplemental Information 9 Seed yield per plant

Click here for additional data file.

The authors gratefully acknowledge the Central Laboratory of the Xishuangbanna Tropical Botanical Garden for providing the research facilities.

Additional Information and Declarations

Competing Interests

Author Contributions

Data Availability

The authors declare there are no competing interests.

Ming-Jun Zhang performed the experiments, analyzed the data, prepared figures and/or tables, authored or reviewed drafts of the paper, and approved the final draft.

Qiantang Fu, Huiying He and Mingyong Tang performed the experiments, authored or reviewed drafts of the paper, and approved the final draft.

Mao-Sheng Chen analyzed the data, authored or reviewed drafts of the paper, and approved the final draft.

Jun Ni performed the experiments, prepared figures and/or tables, authored or reviewed drafts of the paper, and approved the final draft.

Yan-Bin Tao and Zeng-Fu Xu conceived and designed the experiments, analyzed the data, prepared figures and/or tables, authored or reviewed drafts of the paper, and approved the final draft.

The following information was supplied regarding data availability:

The qRT-PCR primers, JcBSP1 expression in wild-type (WT) and transgenic Arabidopsis lines and raw measurements are available in the Supplementary Files.

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
