# Peer review of "Characterization of the bark storage protein gene (JcBSP) family in the perennial woody plant Jatropha curcas and the function of JcBSP1 in Arabidopsis thaliana"

_PeerJ, doi:10.7717/peerj.12938_

## Round 0.1 · original submission · Minor Revisions

Overall, both reviewers considered that this was an interesting manuscript. Both of them also brought up some good suggestions to improve this manuscript.

·

Basic reporting

This report is well written with professional English used throughout. It is well structured and self-contained.

Experimental design

Research questions were well defined, relevant & meaningful. Experiment designs were appropriate. Methods and materials were sufficiently described.

Validity of the findings

Meaningful replications were included to support the findings, which are generally robust and statistically sound.
I commend the authors for the extensive and pioneer work on the Jatropha bark storage protein that may play an important role in nitrogen cycling in this important biofuel crop.

Additional comments

The reviewer has some comments/suggestions for further improvement:
1. Creating a better link to Jatropha in the tropical climate: The authors investigated Jatropha defoliate in autumn and dormancy in wintertime somewhere in China. Jatropha is mostly grown in the tropics as a perennial that sheds off leaves during drought periods. It would be important to link the season change-related protein-related mobilization to possible drought-related protein mobilization.
2. JcBSP1 has a seasonal RNA transcript pattern consistent with season-related total protein concentration change. Authors suggest that JcBP1 might play an important role in seasonal nitrogen cycling. The conclusion will be better supported with the quantification of JcBP1 in the total protein. A high percentage would better support the suggested ‘important role’.
3. Among the 21 transgenic Arabidopsis lines for over-expression JcBSP1 under the constitutive strong 35S promoter, two of them showed higher JcBSP1 transcripts and larger rosette leaves and flowers, also larger seeds. These phenotypes suggested the involvement of JcBSP1 in plant growth and development, not exactly expected for a balk storage protein. It is necessary to exclude the possibility of pleiotropic phenotypes caused by insertion mutagenesis or multiple transgene insertions. Some molecular characterization of transgene copy number and insertion sites for these two lines would be very helpful.

Reviewer 2 ·

Basic reporting

In the paper " Characterization of the bark storage protein gene (JcBSP) family in the perennial woody plant Jatrophacurcas and the function of JcBSP1 in Arabidopsis thaliana", the authors identified six members of JcBSP gene family in J. curcas genome. Thepatterns, seasonal changes, and responses to nitrogen treatment in gene expression of JcBSPs were detected by qRT-PCR. Moreover, JcBSP1 was overexpressed expressed in Arabidopsis resulted in a phenotype of enlarged rosette leaves, flowers, and seeds, and significantly increased the seed weight and yield in transgenic plants. This study laid the foundation for further research on the function of BSP genes in the plant growth and development.

(1)In this manuscript, the JcBSP1 from Jatrophacurcas was expressed in Arabidopsis to verify JcBSP1 function. So, the authors described that “JcBSP1overexpression in Arabidopsis…..”, which might be not accurate.

(2)Is there any homologous genes of JcBSP1 in Arabidopsis thaliana? Does the authors checked that or any literature reported?

(3)In Figure 7B, the X-axis is better simplified as “……(week)”, and uniform the unit of Y-axis as in Figure 9 B and C.

(4)Standard deviations in Figure 8B about qRT-PCR is very big, please more repeats to confirm these results.

Experimental design

This study was well-designed, and the experiments were performed properly.

Validity of the findings

The article is well-drafted, and include sufficient literature references.

---

## Round 0.2 · accepted · Accept

Authors have addressed the concerns brought up by the reviewers.

The Section Editor noted that the authors need to indicate what the error bars represent in Figure 4.

·

Basic reporting

This report is well written with professional English used throughout. It is well structured and self-contained.

Experimental design

Research questions were well defined, relevant & meaningful. Experiment designs were appropriate. Methods and materials were sufficiently described.

Validity of the findings

Meaningful replications were included to support the findings, which are generally robust and statistically sound.
I commend the authors for the extensive and pioneer work on the Jatropha bark storage protein that may play an important role in nitrogen cycling in this important biofuel crop.

Additional comments

The reviewer gave some comments/suggestions for further improvement after the 1st review. These questions are well addressed in the revision.

Reviewer 2 ·

Basic reporting

The authors addressed all the questions that I raised. I have no more questions now.

Experimental design

The authors addressed all the questions that I raised. I have no more questions now.

Validity of the findings

The authors addressed all the questions that I raised. I have no more questions now.